# mTOR in the Development of Hypoxic Pulmonary Hypertension Associated with Cardiometabolic Risk Factors

**DOI:** 10.3390/ijms252011023

**Published:** 2024-10-14

**Authors:** Karen Flores, Carlo Almeida, Karem Arriaza, Eduardo Pena, Samia El Alam

**Affiliations:** High Altitude Medicine Research Center (CEIMA), Arturo Prat University, Iquique 1110939, Chile; kfloresu@unap.cl (K.F.); calmeida.biotec@gmail.com (C.A.); eduardopena@unap.cl (E.P.); selalam@unap.cl (S.E.A.)

**Keywords:** hypoxic pulmonary hypertension, mTOR, cardiometabolic risk factors

## Abstract

The pathophysiology of pulmonary hypertension is complex and multifactorial. It is a disease characterized by increased pulmonary vascular resistance at the level due to sustained vasoconstriction and remodeling of the pulmonary arteries, which triggers an increase in the mean pulmonary artery pressure and subsequent right ventricular hypertrophy, which in some cases can cause right heart failure. Hypoxic pulmonary hypertension (HPH) is currently classified into Group 3 of the five different groups of pulmonary hypertensions, which are determined according to the cause of the disease. HPH mainly develops as a product of lung diseases, among the most prevalent causes of obstructive sleep apnea (OSA), chronic obstructive pulmonary disease (COPD), or hypobaric hypoxia due to exposure to high altitudes. Additionally, cardiometabolic risk factors converge on molecular mechanisms involving overactivation of the mammalian target of rapamycin (mTOR), which correspond to a central axis in the development of HPH. The aim of this review is to summarize the role of mTOR in the development of HPH associated with metabolic risk factors and its therapeutic alternatives, which will be discussed in this review.

## 1. Introduction

Pulmonary hypertension is a multifactorial pathology characterized by a sustained elevation in mean pulmonary artery pressure at rest (>20 mmHg) [1] resulting from vasoconstriction and remodeling of the pulmonary artery [2]. At present, therapies mainly provide symptomatic relief by addressing imbalances in vasoactive factors. Novel treatments, including inhaled tyrosine kinase inhibitors and suppressors of activin receptor type IIA, are showing promise, but long-term safety remains uncertain. Future PAH management will focus on right ventricular function and personalized medicine [3]. Pulmonary hypertension is classified into five groups: Group 1 = pulmonary arterial hypertension; 2 = pulmonary hypertension caused by left heart disease; 3 = pulmonary hypertension caused by lung disease and/or hypoxia; 4 = pulmonary hypertension caused by pulmonary artery obstruction; and 5 = pulmonary hypertension with unclear and/or multifactorial mechanisms. Notably, in Group 3, pulmonary hypertension is also called HPH [1].

Among the respiratory diseases associated with HPH are COPD and OSA [4]. In addition, long-term exposure to high altitudes may be an environmental condition for the development of HPH, also called high-altitude pulmonary hypertension (HAPH) [1,5]. During this exposure, HPH is characterized by a mean arterial pressure (>30 mmHg) [6].

These alterations in the pathophysiological mechanisms of HPH have also been described to be aggravated by other causes, such as cardiometabolic risk factors, which induce the proliferative and inflammatory state of the cells [6,7,8]. The importance of the proliferative protein mTOR, which is activated under conditions of hypoxia, inflammation and metabolic alterations contributing to HPH, is highlighted [9]. In recent decades, mTOR has been postulated as a therapeutic target for the treatment of HPH associated with cardiometabolic risk factors. This review focuses on discussing the mechanisms associated with these previously described pathologies.

## 2. mTOR

mTOR is a serine/threonine protein kinase of approximately 288 kDa consisting of 2550 amino acids and belongs to the phosphoinositide kinase-related kinase (PIKK) family [10]. This protein presents two isoforms that interact with other specific adaptor proteins to form two macromolecular complexes, namely, mTOR complex 1 (mTORC1) and mTOR complex 2 (mTORC2) [11,12]. The mTORC1 complex consists of a subunit called regulatory-associated protein of mTOR (RAPTOR), a proline-rich Akt substrate of 40 kDa subunit (PRAS40), the target of rapamycin complex subunit LST8 (mLST8), and the DEP domain-containing mTOR-interacting protein (DEPTOR) [13,14]. The RAPTOR subunit and mLST8, together with mTOR, represent the core components of mTORC1. RAPTOR is required for substrate recruitment and subcellular localization of the complex, whereas mLST8 is associated with its catalytic domain, which is required for kinase activity [15]. On the other hand, DEPTOR and PRAS40 represent the inhibitory subunits. The mTORC2 complex is also composed of DEPTOR and mLST8, in addition to the rapamycin-insensitive mTOR-associated protein (RICTOR) and regulatory assembly subunits such as mammalian stress-activated protein kinase (mSin), interacting proteins and proteins observed with RICTOR (PROTOR) [16,17].

Under physiological conditions, mTOR is activated by stimuli such as energy metabolism and growth factors, performing essential functions in the cell. Thus, mTORC1 promotes proliferation through protein synthesis, cell metabolism, lipid synthesis, and the biogenesis of autophagosomes and lysosomes [18]. mTORC2 participates in the regulation of survival, apoptosis, gluconeogenesis, hyperglycemia, ion transport and cytoskeleton formation [11,19].

However, under conditions of cellular stress, such as hypoxia and metabolic alterations, mTOR is overactivated, promoting signals associated with cell growth and proliferation pathways, which leads to the development of various pathologies, such as metabolic syndrome, cancer and cardiovascular diseases, including pulmonary hypertension [10,20].

## 3. Hypoxic Pulmonary Hypertension and mTOR

Prolonged alveolar hypoxia is a consequence of various pathologies and stressors associated with respiratory control and pulmonary function, which lead to HPH [21,22,23]. HPH is characterized by increased pulmonary artery pressure as a consequence of hypoxic pulmonary vasoconstriction (HPV) and pulmonary artery remodeling [24,25], associated with endothelial dysfunction and the proliferation of pulmonary artery smooth muscle cells (PASMCs) under hypoxic conditions [26,27,28], which can lead to right ventricular hypertrophy that, in some cases, can lead to heart failure [29]. The main mechanisms involved in the development of HPH are oxidative stress and inflammatory factors [30,31]. Among the molecules involved in the activation of these mechanisms are reactive oxygen species (ROS), such as superoxide anion (O_2_^−^), which trigger HPV determinant pathways, increasing intracellular calcium and subsequent contraction of PASMCs, decreasing the bioavailability of the vasodilator nitric oxide, unbalancing the vasoconstrictor and vasodilator state, and activating proliferative pathways through growth and inflammatory factors, such as platelet-derived growth Factor B (PDGF), transforming growth factor-β (TGF-β), and inflammatory factors, such as interleukin-6 (IL-6) [32,33,34,35]. In response to hypoxia, all these molecules are related to hypoxia-inducible transcription Factor 1-α (HIF1-α), which can transcribe more than 200 genes related to the activation of metabolic and proliferative pathways to adapt to this condition [36,37].

In patients with HPH, the mTOR pathway promotes proliferation through increased growth factors such as PDGF, epidermal growth factor (EGF) and fibroblast growth factor (FGF) and directly through protein synthesis, which promotes the proliferation of PASMCs [38], pulmonary artery endothelial cells (PAECs) and adventitia [39,40,41,42]. In addition, the overexpression of protein kinase B (PKB), also known as Akt, one of the main upstream kinases that activates mTOR, has been described. In turn, studies of HPH due to exposure to chronic hypoxia (at 10% O_2_ for the subsequent 4 weeks) show that the Akt/mTOR axis is regulated by the overexpression of proteins with proliferative functions, such as c-Src proto-oncogene (c-Src) and phosphoinositide 3-kinase (PI3K) [43,44,45,46], which are among the most described.

The following section details the most prevalent respiratory diseases leading to the development of HPH and the regulation of mTOR and its pathways in these pathologies.

## 4. Chronic Obstructive Pulmonary Disease (COPD)

COPD is a multifactorial type of lung disease that makes breathing difficult, and causes progressive airflow obstruction, triggering chronic hypoxia at the pulmonary and systemic levels. Genetic and environmental factors, including lung irritation, aging, exposure to combustion and air pollution or smoking, are associated with the development of COPD [47]. There were 3.2 million deaths due to chronic obstructive pulmonary disease (COPD), being the seventh cause of death in the world [48]. An important consequence of COPD is HPH, where mTOR has been shown to play an important role [42].

An important mTOR pathway is pulmonary cellular senescence in COPD. In aging, COPD is manifested by an increased level of oxidative stress and accelerated lung cellular senescence activation through activation of the PI3K/Akt/mTOR pathway [49,50,51], as corroborated by studies in a murine model, where overactivation of mTOR stimulates lung cell senescence, with the rapid development of lung emphysema, pulmonary hypertension, and inflammation [52]. Derived research has demonstrated that erythromycin ameliorates oxidative stress-induced cellular senescence via the PI3K-mTOR signaling pathway in COPD [53].

Another important finding involves the mTOR/cell division cycle 42 (Cdc42) pathway, whose activation is related to the deficiency of phagocytosis of alveolar macrophages associated with COPD pathology [54].

## 5. Obstructive Sleep Apnea (OSA)

OSA is a clinical sleep disorder and is associated with increased morbidity and mortality from cardiovascular disease [55]. This pathology is characterized by episodes of interrupted breathing for short periods during nocturnal sleep, leading to intermittent systemic hypoxia [56]. The cyclic pattern of hypoxia and reoxygenation reduces nitric oxide bioavailability and increases oxidative stress [57,58]. OSA is strongly associated with pulmonary hypertension, with a reported prevalence of up to 40% to 80% among diagnosed patients [59]. Intermittent hypoxia has been shown to promote the activation of HIF1α and mTOR, both of which promote cell proliferation in this disease. In addition, mTOR activation by hypoxia is associated with circadian rhythm dysregulation, which increases the expression of mothers against decapentaplegic homolog 4 (SMAD4). SMAD4 is a crucial protein in the regulation of the TGF-β pathway related to angiogenesis and cell proliferation, among other processes. Therefore, elevated plasma SMAD4 levels are related to several cardiometabolic complications, such as diabetes, dyslipidemia and pulmonary hypertension, in patients with OSA [60]. On the other hand, activation of the Akt/mTOR pathway to promote the development of HPH has also been described in this disease [61].

Another pathway described is that of Sestrin2, which has been found to be elevated in the plasma and urine of patients with OSA and is considered diagnostic of the disease. Sestrin2 is overexpressed under hypoxic conditions through HIF1-α as a protective mechanism to reduce ROS levels and anti-proliferative effects through the activation of the AMP-activated protein kinase (AMPK) pathway, one of the main inhibitory kinases of mTOR and vice versa [62,63,64], where AMPK and mTOR are inversely regulated [65]. Another study in OSA patients also revealed the possible role of mTOR in the development of pulmonary hypertension through alteration of the gut microbiota, which is associated with sleep disruption [66].

## 6. Development of HPH Due to Exposure to Altitude

Living at high altitudes, such as in mountainous geographical areas of the Andes, Tibet and Ethiopian highlands, as is the case for a large part of the world’s population, results in exposure to chronic hypobaric hypoxia [67]. Under this condition of chronic hypobaric hypoxia, overactivation of mTOR has been described in studies in murine models, simulating 5000 m of altitude, which is attributed to the development of pulmonary hypertension [68]. Similarly, in other studies simulating an altitude of 4600 m, overactivation of mTOR, which is associated with increased development of right ventricular hypertrophy, has been reported [69]. In accordance with previous reports, HPH was induced in a murine model via hypobaric hypoxia at a simulated altitude of 4500 m above sea level, where eight core genes were identified, with mTOR being key in the development of HPH and associated with immune system deregulation. This dysregulation results in the release of macrophages, mast cells, CD8+ cytotoxic T lymphocytes and memory CD4+ lymphocytes, which are involved in PASMCs inflammation and the subsequent development of HPH [70,71]. Importantly, studies on chronic hypoxia have focused mostly on signaling pathways related to oxidative stress, inflammation and proliferation. In models of chronic hypoxia, the latter signaling pathway promotes the development of HPH through the activation of PI3K/Akt/mTORC1 kinases, a product of cell proliferation in PASMCs [41,42,72]. Another important pathway associated with mTOR and HPH under conditions of chronic hypobaric hypoxia is the AMPK protein, which has been shown to have cardioprotective functions by inhibiting mTOR. Therefore, AMPK activation could mitigate the development of HPH [41,69,73].

Another type of high-altitude exposure is called chronic intermittent hypobaric hypoxia (CIHH), which is a characteristic of intermittent exposure to high altitudes [74,75]. Exposure to CIHH is a condition to which many people may be exposed in a variety of situations, including work or sports. For example, in northern Chile, there is a condition of high-altitude shift system work, in which for a long period, from 6 months to years, they are subjected to intermittent exposure to high altitudes, according to the “Technical Guide on occupational exposure to chronic intermittent hypobaria due to high altitude” [8,76]. Under these hypoxic conditions, the mTOR signaling pathway associated with cardiovascular damage in the right circuit is activated. On the basis of these findings, in murine models, which simulate an intermittent system, i.e., two days at high altitude and 2 days at sea level, for a period of 30 days, simulating an altitude of 4600 m, they reported that the mTOR pathway involved in the development of right ventricular hypertrophy is body weight dependent, where weight reduction helps to reduce hypertrophy by inhibiting mTOR through the AMPK pathway [69]. In addition, coincident studies indicate that the inhibition of mTOR by AMPK exerts a protective effect by decreasing the development of pulmonary hypertension [77]; thus, under this CIHH condition, the activation of mTOR inhibitory pathways is also a promising cardiovascular therapeutic target [78,79].

The mTOR signaling pathway is a central axis in the development of different respiratory and pulmonary diseases and is affected by hypoxia in a dose-dependent (chronic or intermittent) manner. In addition, the existence of key cardiometabolic risk factors that may aggravate the development of HPH is described below.

## 7. Cardiometabolic Risk Factors

The World Health Organization considers smoking, a sedentary lifestyle, a diet rich in cholesterol and saturated fat, dyslipidemia, arterial hypertension, diabetes, and obesity as the main cardiometabolic risk factors [80,81]. Studies have shown that metabolic disturbances exacerbate the development of pulmonary hypertension caused by respiratory diseases [82,83,84,85]. On the other hand, cardiometabolic risk factors are potent activators of mTOR. In this sense, the pathways in which mTOR acts have been postulated to be one of the molecular mechanisms linking the development of pulmonary hypertension with metabolic syndrome [9].

In this section, the cardiometabolic risk factors that most influence the mTOR pathway are mentioned.

### 7.1. Overweight and Obesity

The prevalence of overweight and obesity is associated with increased cardiovascular risk linked with weight gain [86]. For those who suffer from HPH derived from respiratory pathologies, overweight and obesity have been established as aggravating risk factors leading to increased HPH [87,88]. For example, in cardiopulmonary diseases related to high-altitude exposure, Honigman et al. suggested that obese individuals are more susceptible to developing HAPH, as well as other diseases contributing to high-altitude cardiovascular impairment, such as excessive erythrocytosis [6,87,89,90,91]. These findings are also supported by studies in people with high-altitude shift system work, where overweight and obesity, especially increased waist circumference, are associated with the development of HPH [8]. Likewise, obesity is important for increasing middle pulmonary artery pressure (mPAP) levels in OSA and COPD patients [92]. The common factor is hypoventilation resulting from increased body and visceral fat, which are associated with decreased lung function and increased hypoxemia, leading to increased development of HPH [90,93]. At the molecular level, increased hypoxemia under obese conditions activates systemic inflammatory pathways through the activation of HIF-1α in hypertrophied obese adipose tissue [94] and through the inhibition of adipocytokines [95]. Thus, another pathway that is altered in obese individuals and contributes to the development of pulmonary hypertension is adiponectin. Adiponectin is a hormone synthesized by adipose tissue that has a protective effect on the pulmonary vasculature by inhibiting inflammatory and proliferative pathways such as the mTOR and nuclear factor kappa-light-chain-enhancer of activated B cells (NF-κβ) pathways [78]. Obesity has been shown to promote decreased adiponectin levels through an increase in the levels of inflammatory factors such as IL-6 and tumor necrosis factor-alpha (TNF-α) in adipocytes [96]. In addition, perivascular fat produced by obesity is the main cause of vascular resistance to adiponectin, leading to AMPK inhibition and the contribution of pulmonary hypertension via mTOR [96,97,98]. Thus, adiponectin represents a potential link between obesity and HPH [96].

### 7.2. Insulin Resistance

Insulin resistance plays a crucial role in pathologies such as type II diabetes and obesity. Under conditions of metabolic syndrome, the inflammatory response of perivascular fat may contribute to the pathogenesis of vascular insulin resistance [99]. Studies have shown that insulin resistance is closely related to the development of pulmonary hypertension and could be considered an associated risk factor [100]. Similarly, in patients with pulmonary hypertension, a high prevalence of insulin resistance has been observed [101]. In the case of HPH due to COPD, OSA and exposure to high altitude, several researchers have described the importance of insulin resistance and its contribution to the increase in HPH [8,102,103]. Thus, studies in subjects exposed to high-altitude CIHH (4800 m) presented a significant prevalence of insulin resistance, with a strong association with HAPH [8].

The association between insulin resistance and HAPH may be due to the decreased vasodilator action of insulin on the pulmonary artery endothelium, promoting endothelial dysfunction and vasoconstriction [104,105]. Under physiological conditions, the main function of insulin at the vascular level is to induce the production of nitric oxide, the main vascular vasodilator, through the activation of the PI3K/Akt pathway by endothelial nitric oxide synthase (eNOS). In the endothelium, this mechanism is initiated by activation of the insulin receptor (IR), followed by phosphorylation of the insulin receptor substrate (IRS-/IRS-2) [106,107]. IRS-1 is required for endothelial nitric oxide production [106], whereas IRS-2 is involved mainly in protection against vascular damage [108]. Therefore, decreased nitric oxide bioavailability may be one of the mechanisms by which insulin resistance contributes to endothelial dysfunction and HPH [104,109,110,111].

Regarding the role of mTOR in insulin resistance, it has been proposed that hyperinsulinemia and chronic activation of Akt signaling result in selective activation of the PI3K/Akt pathway [112,113,114], inducing overactivation of mTOR, which promotes cell proliferation and, furthermore, vascular insulin resistance via S6K1 (p70S6 kinase) and growth factor receptor-bound protein 10 (Grb10) through uncoupling of IR and PI3K signaling. Akt directly inhibits tuberous sclerosis complex 1/2 (TSC1/2) proteins, which in turn blocks the homologous Ras homolog enriched in the brain (RHEB), leading to overactivation of mTORC1 [115,116,117,118,119,120,121]. In addition, the mTOR/S6K1 pathway is overexpressed by hypoxic stress, excessive carbohydrate and saturated fat consumption, as it occurs in HPH under obese conditions [41,42,122,123]. These findings support the findings of animal model studies, which demonstrated that insulin resistance coupled with hypoxia, as well as macrophage recruitment, induced an increase in mPAP [124].

On the other hand, the mTORC2 isoform also responds to PI3K stimulation via insulin and acts synergistically with Akt overactivation, promoting proliferation [125,126]. Therefore, evidence indicates that mTOR activation is dependent on Akt activation, which inhibits vasodilation and promotes proliferative pathways that contribute to the development of HPH [119,127,128,129,130,131].

In addition, AMPK, a kinase with antiproliferative and HPH mitigating effects that inhibits mTOR [132], is inhibited by overweight and hyperinsulinemia, with overactivation of mTOR being observed under these conditions [69].

### 7.3. Dyslipidemia

Dyslipidemia is a cardiovascular risk factor and an important risk factor for the development of HPH resulting from excess perivascular fat [133,134,135]. In people exposed to high altitude, there is a high prevalence of dyslipidemia, which is strongly associated with HPH [90,136,137]. The same finding was reported in an animal model exposed to hypobaric hypoxia [138]. Similarly, in patients with COPD and OSA, there is a high prevalence of dyslipidemia [139,140].

The reasons for this dyslipidemia are attributed to several pathways activated by hypoxemia, such as increased adipose tissue lipolysis, increased triglyceride synthesis, the secretion of hepatic lipoproteins such as VLDL, and the inhibition of lipoprotein clearance [141,142]. HIF is one of the central molecules involved in activating fat pathways under hypoxic conditions. In rats exposed to intermittent hypoxia, HIF-1 was shown to stimulate the activity of sterol regulatory element-binding protein-1 (SREBP-1), which regulates the enzyme stearoyl-CoA desaturase-1 (SCD-1), which induces triglyceride synthesis [143]. Another study demonstrated de novo triglyceride synthesis under conditions of chronic hypobaric hypoxia, where an increase in SCD-1 elevated hepatic triglyceride levels [144]. In this context, mTOR is also activated by dyslipidemia under hypoxic conditions [145,146], activating the mTOR/S6K pathway and promoting endothelial dysfunction [123]. mTOR also plays an important role in the regulation of lipid biosynthesis through mTORC1/SREBP regulation, modulating the expression of genes associated with cholesterol and fatty acid biosynthesis [147]. SREBP is activated by mTORC1 through ribosomal protein S6 kinase beta-1 (S6K1), which phosphorylates and activates SREBP cleavage-activating protein (SCAP). This leads to the translocation of the active form of SREBP to the nucleus and positive regulation of genes involved in lipid synthesis [148]. Overactivation of SREBP-1 induces the expression of proinflammatory cytokines, which in turn activate SREBP-1, including TNF, IL-1β and interferon-γ [149], resulting in positive feedback to accelerate lipid accumulation and inflammation. Therefore, inhibition of this pathway is key to the treatment of dyslipidemia and pathologies such as hypertension [150,151,152,153,154,155,156,157,158].

The mTOR/S6K1 pathway, which promotes metabolic and cardiovascular abnormalities, is the pathway that is most affected by the effects of hypoxia and metabolic syndrome [41,42,122,123,146].

## 8. Therapeutic Targets That Inhibit the mTOR Signaling Pathway

mTOR is considered a potentially attractive therapeutic target for treating disorders associated with cardiac hypertrophy and HPH [159]. In the following, different direct and indirect mTOR inhibitor treatments for this pathology will be given.

### 8.1. Pharmacological Treatment

Pharmacological experiments in animal models support that mTOR inhibition can prevent the development of pulmonary hypertension. There are currently no pharmacological compounds available that are able to effectively inhibit mTORC1 or mTORC2 [38]. In this section, we report on several direct and indirect mTOR antagonist drugs.

Metformin: Metformin has been designated as one of the most important first-line therapeutic agents in the treatment of type II diabetes mellitus and insulin resistance [160]. Its main mechanism of action is to inhibit mitochondrial ATP synthesis, reduce cellular energy and thus act on AMPK activation [161,162]. Metformin-activated AMPK directly inhibits mTOR and indirectly inhibits it through the TSC1/2 complex [41,163,164].

AICAr: 5-aminoimidazole-4-carboxamide ribonucleoside (AICAr) is an AMP analog; thus, its mechanism of action is regulated through AMP/ATP upregulation, and it is thus a potent activator of AMPK in various tissues and cells in vivo and in vitro [165]. AICAr has been reported to activate AMPK, which inhibits mTORC1 but activates mTORC2. Inhibition of mTORC1 requires phosphorylation of the Raptor subunit [166], whereas activation of mTORC2 by AMPK has an important role in controlling glucose metabolism [167]. Thus, the AMPK/mTORC2 pathway promotes glucose homeostasis and changes the understanding of the complex relationship between AMPK and mTOR and their roles in health and disease.

Rapamycin: Rapamycin is a very potent antiproliferative drug that directly inhibits mTOR [39]. It has also been shown to block the PI3K/Akt pathway, thereby inhibiting mitogen-induced proliferation [168,169].

### 8.2. Phytopharmaceuticals

Multiple natural medicines derived from traditional medicines that possess pharmacological activities have been tested. During the past few years, approximately 25 types of natural medicines have been shown to protect against hypoxia-induced pulmonary hypertension injury [170]. Some of the most relevant phytopharmaceuticals for mTOR inhibition are described below.

Curcumin: Curcumin (diferuloylmethane) is a molecule isolated mainly from *Curcuma longa* [113]. In cell culture and animal models, curcumin has been shown to have antiproliferative activity and is a potential therapeutic agent for diabetes and cardiovascular and pulmonary diseases [171]. A study by Johnson reported that, in cell culture, curcumin has antiproliferative activity and can decrease the activity of mTORC1, decreasing the phosphorylation of downstream effects such as S6K1 and 4E-BP1. In addition, curcumin administration decreased the expression of the Raptor and Rictor components of the mTORC1-C2 complex [172].

Resveratrol: Resveratrol is a polyphenolic compound derived mainly from Vitis vinifera and Reynoutria japonica that has antifibrotic, anti-inflammatory and cardiovascular protective effects [173]. A study by Peng et al. demonstrated in animal models exposed to chronic intermittent hypoxia that intragastric administration of resveratrol significantly improved cardiac function, oxidative stress and autophagy activation through inhibition of PI3K/Akt/mTOR [174].

### 8.3. Gene Therapy

MicroRNAs are small endogenous noncoding RNA sequences that regulate gene expression and influence various biological processes [175]. Changes in miRNA expression levels contribute to various cardiovascular disorders and play a key role in the pathogenesis of pulmonary hypertension. There are different types of microRNAs with antiproliferative properties associated with the development of pulmonary hypertension, some relevant ones will be discussed below.

miR-124: The nuclear factor of activated T cells (NFAT) signaling pathway is crucial for cell proliferation and pulmonary hypertension. Apparently, miR-124 plays a key role by inhibiting NFAT-dependent interleukin 2 (IL-2) transcription through targeting NFATc1, calmodulin-binding transcription activator 1 (CAMTA1) and polypyrimidine tract-binding protein 1 (PTBP1). miR-124 is down-regulated in hypoxia-induced human pulmonary arterial smooth muscle cells (HPASMCs) and mice lungs, with its overexpression shown to inhibit HPASMC proliferation [176]. Similarly, miR-124 levels are reduced in fibroblasts from experimental pulmonary hypertension models and pulmonary hypertension patients. This reduction leads to increased fibroblast proliferation and migration, while overexpression of miR-124 blocks these effects [177]. Thus, miR-124’s antiproliferative properties could be valuable for developing pulmonary hypertension treatments.

miR-424/503: The expression of Apelin (APLN) is reduced in PAECs from PH patients, leading to the downregulation of miR-424/503, which in turn promotes the expression of fibroblast growth factor 2 (FGF2) and fibroblast growth factor receptor (FGFR1) in PAECs [178,179]. miR-424/503 is down-regulated in pulmonary hypertension patients, causing up-regulation of FGF2/FGFR1. In experimental pulmonary hypertension models, sustained elevation of miR-424/503 via intranasal lentiviral delivery prevented right ventricular systolic pressure elevation and reversed vascular remodeling. This suggests that restoring miR-424/503 function could offer clinical benefits in pulmonary hypertension [179].

miR-98: research has shown that miR-98 is down-regulated in hypoxia-induced PAECs, lungs of mice exposed to Sugen5416/hypoxia, and PAECs from pulmonary hypertension patients. Also, miR-98 is positively regulated by peroxisome proliferator activated receptor gamma (PPARγ) and directly targets endothelin 1 (ET-1), which plays a role in PAEC proliferation [180].

miR-193: It has been previously described that miR-193 directly targets the insulin-like growth factor-1 receptor (IGF1R), and its overexpression inhibits PASMC proliferation in pulmonary hypertension. Additionally, intratracheal administration of miR-193 mimics alleviated pulmonary hypertension symptoms in monocrotaline and hypoxia models [181].

miR-150: Several studies have shown that miR-150 is an important regulator of B-cell differentiation, proliferation, metabolism and apoptosis [182]. A study by Ying et al. reported that the administration and subsequent overexpression of miR-150 inhibited the Akt/mTOR signaling pathway, alleviating PASMC lung remodeling in rats with HPH [183].

miR-100: miR-100 has been reported to be associated with cell proliferation and apoptosis [184]. Wang et al. studied the expression of miR-100 in PASMC tissues and administered miR-100 mimics and reported that their effect was positive in suppressing cell proliferation under hypoxic conditions. Therefore, the link between miR-100 and mTOR has significant implications for PASMC proliferation, remodeling and pulmonary hypertension [155,156,157]. Below is a summary of the role of miRNAs on pulmonary hypertension (Table 1).

The pathways involved in the development of HPH associated with metabolic risk factors and its therapeutic alternatives, are summarized in Figure 1.

## 9. Conclusions

This review highlights that the development of pulmonary hypertension (PH) is linked to the activation of the mTOR pathway, which contributes to vascular remodeling by promoting the proliferation of pulmonary artery smooth muscle and endothelial cells. Risk factors like obesity, dyslipidemia, and insulinemia can exacerbate this process, worsening PH progression. Addressing both mTOR signaling and these metabolic risk factors could enhance management strategies for hypoxic pulmonary hypertension. Potential therapeutic approaches include pharmacological treatments like Rapamycin (an mTOR inhibitor), phytotherapy with Curcumin and Resveratrol, and gene therapy using miRNAs, all showing promise in mitigating smooth muscle cell proliferation. This research opens new avenues for understanding the interplay between hypoxia, metabolic disorders, and pulmonary hypertension.

## Figures and Tables

**Figure 1 ijms-25-11023-f001:**
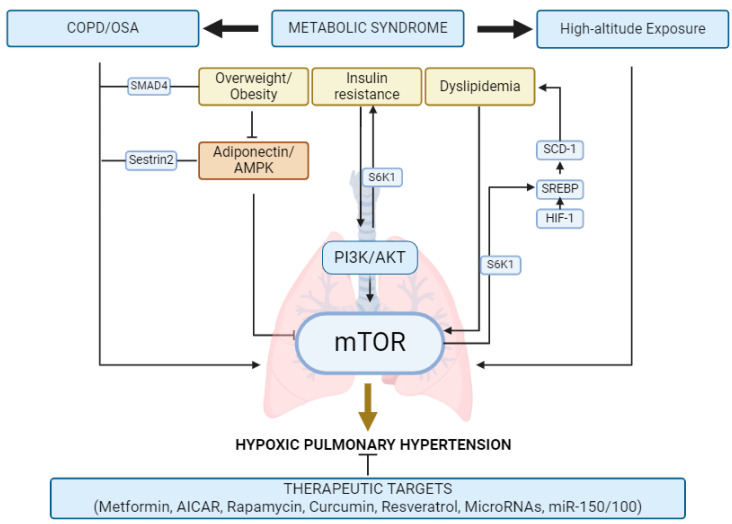
Development of HPH associated with metabolic risk factors and its therapeutic alternatives. COPD: chronic obstructive pulmonary disease; OSA: obstructive sleep apnea; SMAD4: mothers against decapentaplegic homolog 4; AMPK: AMP-activated protein kinase; S6K1: p70S6 kinase; PI3K: phosphoinositide 3-kinase; AKT: protein kinase B (PKB); mTOR: mammalian target of rapamycin; SCD-1: stearoyl-CoA desaturase-1; SREBP: sterol regulatory element-binding protein; and HIF-1: Hypoxia-inducible factor 1. Created using BioRender (https://www.biorender.com/) 5 September 2024.

**Table 1 ijms-25-11023-t001:** Commonly dysregulated miRNAs under hypoxia in pulmonary hypertension.

miRNA	Model of Research	Effects in Cells or Tissues	Targets for Pulmonary Hypertension	References
miRNA-124	Human PASMCs and mouse lungs induced by hypoxia	Proliferation ↓ Migration ↓	*NFATc1* *CAMTA1 and PTBP1*	[176,177]
miRNA-424/503	PAECs from PH patients	Proliferation ↓	*FGF2/FGFR1*	[178,179]
miRNA-98	PAECs from PH patients, PAECs under hypoxia and in lung from mice induced by Sugen5416/hypoxia	Proliferation ↓	*ET-1*	[180]
miRNA-193	MCT, hypoxia	Proliferation ↓	*IGF1R*	[181]
miRNA-150	Hypoxia	Lung remodeling Proliferation ↓	Akt/mTOR	[182,183]
miRNA-100	Hypoxia	Proliferation ↓	*mTOR*	[155,157,184]

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
