# Peer review of "mTOR in the Development of Hypoxic Pulmonary Hypertension Associated with Cardiometabolic Risk Factors"

_ijms, 2024, doi:10.3390/ijms252011023_

Round 1
Reviewer 1 Report
Comments and Suggestions for Authors
General concept Comments
1. Title: Clear.
2. Abstract: optimal written. Line 11: Please correct, pulmonary arteries instead pulmonary artery.
3. Introduction: Please update your introduction about pathology of pulmonary hypertension:
https://www.nature.com/articles/s41569-024-01064-4#:~:text=Key%20points.%20Established%20therapies%20for%20pulmonary%20arterial%20hypertension
4. mTOR and development of HPH: Could you mentioned which Burden of Hypoxia is relevant to develop HPH (time of duration, cut-offs are mentioned in literature)?
Line 163: please define ‘long period’ in this study.
Author Response
"Please see the attachment."

Reviewer 2 Report
Comments and Suggestions for Authors
The review titled "mTOR in the Development of Hypoxic Pulmonary Hypertension Associated with Cardiometabolic Risk Factors" summarizes valuable information from the literature on Pulmonary Hypertension, which is of great interest to researchers. Recommendations:
1. The introduction is very brief. Include more general information about the presented pathology.
2. COPD has significant importance and should be discussed in more detail.
3. Add a table listing specific types of microRNAs and their proven implications in pulmonary pathology, especially in diagnostics.
4. Compare routinely used markers with microRNAs.
5. Optional – include information on imaging diagnosis and other diagnostic methods.
6. The conclusion should be more concise.
Author Response
"Please see the attachment."
